# Cytotoxicity and Transcriptomic Analyses of Biogenic Palladium Nanoparticles in Human Ovarian Cancer Cells (SKOV3)

**DOI:** 10.3390/nano9050787

**Published:** 2019-05-22

**Authors:** Sangiliyandi Gurunathan, Muhammad Qasim, Chan Hyeok Park, Muhammad Arsalan Iqbal, Hyunjin Yoo, Jeong Ho Hwang, Sang Jun Uhm, Hyuk Song, Chankyu Park, Youngsok Choi, Jin-Hoi Kim, Kwonho Hong

**Affiliations:** 1Department of Stem Cell and Regenerative Biotechnology and Humanized Pig Center (SRC), Konkuk Institute of Technology, Konkuk University, Seoul 05029, Korea; gsangiliyandi@yahoo.com (S.G.); qasimattock@gmail.com (M.Q.); chanhyeok.park3751@gmail.com (C.H.P.); arsalaniqbal129@gmail.com (M.A.I.); hyunjinyoo7@gmail.com (H.Y.); songh@konkuk.ac.kr (H.S.); chankyu@konkuk.ac.kr (C.P.); choiys3969@konkuk.ac.kr (Y.C.); jhkim541@konkuk.ac.kr (J.-H.K.); 2Animal Model Research Group, Jeonbuk Department of Inhalation Research, Korea Institute of Toxicology, 30 Baekhak1-gil, Jeongeup, Jeollabuk-do 56212, Korea; jeongho.hwang@kitox.re.kr; 3Department of Animal Science and Biotechnology, Sangji Youngseo College, Wonju 26339, Korea; sjuhm@sy.ac.kr

**Keywords:** cytotoxicity, palladium nanoparticles, ovarian cancer, oxidative stress, RNA-Seq analysis

## Abstract

Ovarian cancer incidence continues to increase at an alarming rate. Although various therapeutic approaches exist for ovarian cancer, they have limitations, including undesired side effects. Therefore, nanoparticle (NP)-mediated therapy may be a viable, biocompatible, and suitable alternative. To the best of our knowledge, no comprehensive analysis has been undertaken on the cytotoxicity and cellular pathways involved in ovarian cancer cells, particularly SKOV3 cells. Here, we investigated the effect of palladium NPs (PdNPs) and the molecular mechanisms and cellular pathways involved in ovarian cancer. We assayed cell viability, proliferation, cytotoxicity, oxidative stress, DNA damage, and apoptosis and performed an RNA-Seq analysis. The results showed that PdNPs elicited concentration-dependent decreases in cell viability and proliferation and induced increasing cytotoxicity at increasing concentrations, as determined by leakage of lactate dehydrogenase, increased levels of reactive oxygen species and malondialdehyde, and decreased levels of antioxidants like glutathione and superoxide dismutase. Furthermore, our study revealed that PdNPs induce mitochondrial dysfunction by altering mitochondrial membrane potential, reducing adenosine triphosphate levels, inducing DNA damage, and activating caspase 3, all of which significantly induced apoptosis in SKOV3 cells following PdNPs treatment. Gene ontology (GO) term analysis of PdNPs-exposed SKOV3 cells showed various dysregulated pathways, particularly nucleosome assembly, telomere organization, and rDNA chromatin silencing. When genes downregulated by PdNPs were applied to GO term enrichment analysis, nucleosome assembly was the top-ranked biological pathway. We also provide evidence for an association between PdNPs exposure and multiple layers of epigenetic transcriptional control and establish a molecular basis for NP-mediated apoptosis. These findings provide a foundation, potential targets, and novel insights into the mechanism underlying toxicity and pathways in SKOV3 cells, and open new avenues to identify novel targets for ovarian cancer treatment.

## 1. Introduction

According to the Cancer Statistics Center of the American Cancer Society, ovarian cancer is the fifth most common cause of cancer-related deaths in women [1,2]. Ovarian cancer has the highest mortality rate among gynecological cancers. To reduce the mortality rate, various treatment strategies, including surgery, hormonal therapy, chemotherapy, radiation therapy, immunotherapy, and targeted therapy, are normally applied. Although these conventional therapies have improved patient survival, they also have several limitations [1]. For example, although chemotherapy is a powerful therapeutic approach for the treatment and management of cancer, it also has several undesired side effects and other disadvantages such as non-specificity, drug resistance, and excessive toxic effects. Consequently, it is essential that alternative and suitable therapies are found, such as nanotechnology-mediated cancer therapy for which several roles are known, including cancer diagnosis and treatment, identification of biomarkers, understanding of cancer progression, and finally, development of novel diagnostic and imaging agents [3,4].

Over the past few decades, substantial efforts have been made to synthesize nanoparticles (NPs) of different metals, including silver, gold, zinc, platinum, and palladium, using physical, chemical, and biological methods. All these methods were adopted to maximize NP synthesis due to their remarkable physical and chemical properties that allow for the extensive use of nanomaterials in various purposes, including catalytic, electronic, magnetic, optical, mechanical, and biomedical applications [5,6,7]. Palladium NPs (PdNPs) have attracted considerable interest due to their applications as catalysts in many organic transformations, including various types of carbon–carbon cross-coupling [8], oxidation and reduction reactions, and catalytic conversion [9,10,11]. Conventionally, PdNPs are synthesized through various physical or chemical methods using toxic and hazardous chemicals, like reducing and stabilizing agents. However, several laboratories have recently developed a green chemistry approach for the synthesis of metal NPs [12,13,14,15]. These green chemistry approach-derived NPs exhibit biocompatibility, non-toxicity, and are fast-acting and eco-friendly, in addition to overcoming the disadvantages of conventional physical and wet chemical methods. To date, PdNPs have been synthesized using banana peel extracts [16] and leaf extracts of *Pulicaria glutinosa* [12], *Perilla frutescens* [13], *Evolvulus alsinoides* [14], *Euphorbia granulate* [17], and *Ocimum sanctum* [18]. However, most of these reported plant-based catalysts often have disadvantages, such as the need to purify the synthesized PdNPs using downstream processes, which are time-consuming and associated with the presence of numerous impurities. Consequently, it would be highly desirable to select appropriate purified flavonoids that are suitable for quick bioreduction of Pd salts, aid in multiple catalytic reactions, and afford biological activities [15]. Based on these considerations, we set out to exploit flavonoids for synthesizing PdNPs. Overall, studies related to PdNPs synthesis and their applications in biomedical fields are limited compared to those of silver, gold, and carbon NPs, and to the best of our knowledge, the potential of this flavonoid for synthesizing and functionalizing metal NPs has not been explored. Therefore, the aim of this study was to explore the possibility of using hesperidin for PdNPs synthesis.

Although PdNPs have been utilized in several catalytic applications, their use in biomedical applications is limited. Previous studies have shown that PdNPs elicit significant cytotoxic effects in several human cellular models, including respiratory [11,19], peripheral blood [20], cervical [21], liver [22], ovarian [14], and skin cancer (melanoma) cells [23]. In contrast, Fang et al. used PdNPs in the form of Pd nanosheet-covered hollow mesoporous silica NPs as a platform for chemo-photothermal cancer treatment. Palladium complexes of polyamides containing sulfones showed the highest antibacterial and antifungal potency [24]. Palladium NPs also exhibit toxicity against various types of cancer cells and modulate the release and expression of numerous cytokines [25,26,27]. In human ovarian cancer cells, PdNPs were shown to elicit toxicity by increasing oxidative stress, enhancing caspase 3 activity, and inducing DNA damage [14].

Next-generation sequencing technology is being employed to understand the mechanisms underlying cellular responses and to identify the genes and pathways associated with ovarian cancer. Bioinformatics tools have recently been used to visualize expression data for cellular function. RNA sequencing (RNA-Seq) is a revolutionary tool for transcriptome profiling that employs next-generation sequencing technologies to measure transcript levels with increased precision compared to other approaches [28,29]. For example, RNA-Seq is being extensively used to investigate the mechanisms of drug resistance in cancer and providing insight into the complex mechanisms of resistance to anticancer drugs [30,31]. Although numerous studies have reported the synthesis and characterization of PdNPs, a combination of mechanisms underlying PdNPs-mediated cytotoxicity and identification of cellular pathways impacted by PdNPs has not been investigated. Therefore, we focused on three main objectives: first, to synthesize and characterize PdNPs through a simple and environment-friendly approach using hesperidin. Second, to investigate human ovarian cancer cell responses to PdNPs, and finally, to perform RNA-Seq analysis of differential gene expression in human ovarian cancer cells to determine the underlying molecular mechanisms of cytotoxicity induced by PdNPs. To our knowledge, this is the first report demonstrating cellular responses to, and functional aspects of, PdNPs in SKOV3 cells.

## 2. Materials and Methods

### 2.1. Synthesis and Characterization of PdNPs

Synthesis and characterization of PdNPs were carried out according to a previously described method [14].

### 2.2. Cell Viability and Cell Proliferation

Cell viability was measured using a Cell Counting Kit-8 (CCK-8; CK04-01, Dojindo Laboratories, Kumamoto, Japan). Cell proliferation was determined using BrdU according to the manufacturer’s instructions (Roche). Concentrations of PdNPs showing a 50% reduction in cell viability (i.e., half-maximal inhibitory concentration (IC50 values) were then calculated. RNA-SEQ analysis were carried out with the IC50 value.

### 2.3. Membrane Integrity

The membrane integrity of SKOV3 cells was evaluated using an LDH Cytotoxicity Detection Kit. Briefly, cells were exposed to various PdNPs concentrations for 24 h.

### 2.4. Assessment of Dead-Cell Protease Activity

Dead-cell protease activity was assessed using a previously described method [32].

### 2.5. Determination of Intracellular ROS

The levels of ROS were measured according to a previous method based on intracellular peroxide-dependent oxidation of 2′,7′-dichlorodihydrofluorescein diacetate (DCFH-DA, Molecular Probes, Eugene, OR, USA) to form the fluorescent compound 2′,7′-dichlorofluorescein (DCF) [33].

### 2.6. Measurement of MDA Content and Antioxidant Enzyme Activities

Malondialdehyde content and antioxidant enzyme activities were measured using a previously described method [34].

### 2.7. JC-1 Assay

Changes in mitochondrial membrane potential (MMP) were determined using the cationic fluorescent dye JC-1 (Molecular Probes).

### 2.8. Measurement of ATP Levels

The ATP levels were measured in SKOV3 cells according to the manufacturer’s instructions (catalog number MAK135, Sigma-Aldrich, St. Louis, MO, USA).

### 2.9. Measurement of 8-oxo-dG

8-Oxo-dG (8-oxo-7,8-dihydro-2′-deoxyguanosine) and its analogs were used as biomarkers of oxidative DNA damage and oxidative stress. To evaluate PdNPs-induced oxidative stress in SKOV3 cells, an 8-oxo-dG ELISA assay was utilized as described previously [35], following the manufacturer’s instructions (Trevigen, Gaithersburg, MD, USA).

### 2.10. Measurement of Caspase 3 Activity

Measurement of caspase 3 activity was performed according to a method described previously [14].

### 2.11. Cell Cycle Analysis

Cell cycle analysis was carried out according to Petrarca [19]. Briefly, cells were fixed in 70% cold ethanol, stained with 50 µg/mL propidium iodide in phosphate buffer saline (PBS) buffer containing 200 µg/mL RNase and analyzed using a FACSCalibur flowcytometer equipped with the CellQuest software (version, Becton Dickinson, San Jose, CA, USA).

### 2.12. RNA-Seq and Bioinformatics Analysis

Bioinformatics and pathway analysis were carried out as described earlier. Total RNA was isolated from non-treated control and PdNPs-treated SKOV3 cells. The quantity and quality of the RNA were determined using a Bioanalyzer RNA chip (Agilent, Santa Clara, CA, USA). An RNA-Seq library was produced using the Illumina TruSeq stranded total RNA sample preparation kit in accordance with the manufacturer’s instruction (Illumina, San Diego, CA USA). Approximately 1 µg of total RNA was subjected to PCR amplification for cDNA synthesis, fragmentation, and adaptor ligation. After size selection according to agarose gels, a second PCR amplification was carried out. Sequencing was performed on the NextSeq500 platform (Illumina). After sequencing, the reads were cleaned by removing the adaptor sequences and the quality was determined using the FastQC tool and mapped to the NCBI hg19 human genome using the STAR tool. Using the Cufflink tool, differentially expressed genes (DEGs) were determined at a cutoff value of FPKM > 5 and fold change (FC) > 5. Gene ontology (GO) terms were obtained and plotted using the DAVID (Database for Annotation, Visualization, and Integrated Discovery) and GOplot tools, respectively. The scatter plot was generated in R (v3.3.2).

### 2.13. Pathway Analysis

*Kyoto* Encyclopedia of Genes and Genomes (KEGG) pathway analysis with the DEGs was performed using the Cytoscape software (v3.6.1, Institute of Systems Biology, Seattle, WA, USA) ClueGO (v2.5.1, INSERM, Paris, France) plug-in. Pathways with a *p*-value < 0.05 were considered as statistically significant in the analysis. Gene set enrichment analysis (GSEA) (v3.0, BROAD Institute, Cambridge, MA, USA) was also used to determine biological pathways enriched in the DEGs. Co-expression and physical interaction of the DEGs in the MAPK signaling pathway and oxidative phosphorylation were determined using the GenMANIA (v3.5.0, University of Toronto, Toronto, Ontario, Canada) Cytoscape plug-in.

### 2.14. Statistical Methods

All assays were conducted in triplicate, and the results are presented as the means ± standard deviation. All experimental data were compared using Student’s *t*-test. A *p*-value < 0.05 was considered statistically significant. There was a significant difference in treated cells compared to untreated cells with Student’s *t*-test (* *p* < 0.05).

## 3. Results and Discussion

### 3.1. Synthesis and Characterization of PdNPs

Palladium NPs were synthesized using a simple, easy, eco-friendly method that involved treating a solution of palladium chloride with hesperidin, where a color change indicated PdNPs synthesis. To substantiate successful PdNPs preparation, we characterized the as-prepared PdNPs by UV-Vis spectroscopy. Figure 1A shows the PdNPs UV-vis spectra formed at 40 °C, with reduction of Pd (II) ions to PdNPs mediated by hesperidin. The reference-sample PdCl_2_ showed a peak at 425 nm resulting from absorption of Pd (II) ions. The 425 nm peak was absent for reduced samples and a broad, continuous absorption was observed, indicating the complete reduction of Pd (II) ions to PdNPs. Our results are in agreement with those of earlier publications, including that of Nadagouda and Varma, who reported the green synthesis of silver and palladium NPs using coffee and tea extracts [36].

The successful synthesis of PdNPs was further confirmed by X-ray diffraction (XRD) analysis. The XRD pattern of the PdNPs is shown in Figure 1B. Distinct PdNPs peaks were observed at 2θ, with values of 39.90°, 56.50°, 65.54°, and 83.16° diffracted from the (111), (200), (220), and (222) planes with corresponding *d*-spacing values of 2.3, 1.8, 1.4, and 1.2 Å, respectively (JCPDS card no. 001–1201). The most intense and predominant peak among the PdNPs crystals was observed at 39.9°, corresponding to the (111) planes. The broad peak at 39.9° is the characteristic peak of the (111) indices of Pd (0), which is a face-centered cubic structure. Considering the full width at half maximum (FWHM) of the (111) peak, the mean NP diameter was found to be 10 nm, which agrees with the TEM results. According to the Scherrer equation, the average PdNPs size is approximately 10 nm, which is close to that obtained from the TEM spectra. However, when particle size distribution was measured using dynamic light scattering (DLS), an average hydrodynamic diameter of 30 nm was found, possibly indicating the Brownian motion of the particles in aqueous suspension. Other investigators have also found larger particle sizes using DLS compared to those determined by TEM, with different Pd-based materials [15,37]. Our data were consistent with previous studies that used biological materials such as coffee and tea extracts [36], *Anacardium occidentale* [38], *P. glutinosa* extracts [12], *E. alsinoides* [14], and *Garcinia pedunculata* Roxb [15] as reducing and stabilizing agents for the synthesis of PdNPs.

The fourier-transformed infrared (FTIR) spectra of PdNPs depicted three very prominent peaks at 3330, 1630, and 2120 cm^−1^. The first two peaks could be attributed to O–H stretching and bending of the phenolic hydroxyl groups, whereas the third peak could be due to the carbonyl group of hesperidin (Figure 1C). The absorption peak present at approximately 1630 cm^−1^ is characteristic of C=C stretching in aromatics [12]. The characteristic peak positions of O–H stretching vibration and C–O group vibration at 3330 and 1630 cm^−1^ respectively, were observed because the flavonoid facilitated the reduction and stabilization processes [14,15].

Measurement by DLS is a particularly important technique for determining particle size and size distribution in aqueous suspensions. The cytotoxicity of prepared nanomaterials depends on size, in addition to morphology, composition, surface area, and surface chemistry [39,40]. We therefore performed DLS measurements in aqueous solutions to elucidate the size of the hesperidin-assisted PdNPs, with the results indicating an average size of 10 nm (Figure 1D).

The hydrodynamic size and zeta potential of PdNPs in DMEM media is shown in Table 1. Previously, PdNPs synthesized using *E. alsinoides* exhibited sizes in the range of 1 to 10 nm, with an average size of 5 nm, while PdNPs synthesized using *G. pedunculata* showed an average size of 330 nm, which was larger than usual [15]. Polypyrrole-palladium nanocomposite particles showed an average diameter of 89 nm in an aqueous dispersion [37].

The morphology of hesperidin-assisted PdNPs was analyzed by transmission electron microscopy (TEM). The TEM images revealed that PdNPs are well-dispersed and significantly spherical in shape, with an average size of 10 nm (Figure 1E). The histogram of the TEM images shows ranges between ~4 and 14 nm (Figure 1F). Interestingly, the synthesized particles were found to be well-dispersed and non-aggregated. *Anacardium occidentale* leaf extract produced NPs with a particle size range between 2.5 and 4.5 nm [38]. Palladium NPs also showed a dendritic Pd superstructure in the presence of cetyltrimethylammonium bromide (CTAB) [41]. Particle size depends on the type of reducing agent, temperature, and the concentration of PdCl_2_ used for synthesis. For instance, using coffee and tea extracts, *P. glutinosa* leaf extract, and xanthan gum results in particle sizes of 20–60 nm, 20–25 nm, and 2–12 nm, respectively. Our findings clearly showed that the particle size agrees well with the size obtained from the TEM analysis as well as for the metallic PdNPs synthesized by refluxing Pd acetylacetonate (acac), which was 10 nm [19]. Collectively, our findings suggest that biomolecule-mediated PdNPs synthesis is comparable to that mediated by chemical-assisted synthesis.

### 3.2. PdNPs Inhibit Cell Survival and Proliferation

To investigate the cytotoxic potential of PdNPs, SKOV3 cells were treated with various PdNPs concentrations (1–5 µg/mL), followed by a CCK-8 assay, which measures mitochondrial activity. After 24 h of exposure, mitochondrial activity declined, even at a low concentration of 1 µg/mL and increasing the PdNPs concentration rapidly decreased cell viability (Figure 2A). At the highest tested PdNPs concentration (5 µg/mL), mitochondrial activity showed a significant 90% reduction compared to untreated cells. These results suggest that PdNPs exhibit dose-dependent toxicity against SKOV3 cells, which is in agreement with previous studies that revealed dose-dependent PdNPs cytotoxic effects in primary bronchial epithelial cells (PBECs) and lung carcinoma epithelial cells (A549) [19,42]. PdNPs also showed significant toxicity in peripheral blood [20], as well as in cervical [21], liver [22], ovarian [14], and skin (melanoma) cancer cells [23]. Additionally, bimetallic Pd and platinum NPs markedly affected cell viability in human epithelial cervical cancer HeLa cells [21].

Next, we were interested in determining the potential effect of PdNPs on cell proliferation, one of the critical factors that regulate development. SKOV3 cells were treated with various PdNPs concentrations for 24 h (Figure 2B). After 24 h of treatment, the cells lost their proliferative potential. Appropriate cell proliferation is necessary for functioning and the result indicates that PdNPs have a potential impact on cell proliferation. These results suggest that PdNPs elicit a dose-dependent effect on SKOV3 cells, which is also consistent with the cell viability data. Lavicoli et al. demonstrated that PdNPs dose- and time-dependently inhibited the growth of A549 cells [42]. Moreover, PdNPs were shown to induce progressive cell cycle arrest, with an accumulation of cells in the G0/G1 phase of the cell cycle, suggestive of a potential toxic effect of PdNPs on DNA [20]. Alarifi et al. demonstrated that PdNPs inhibit proliferation of human skin malignant melanoma cells (A375) in a dose- and time-dependent manner through reduction of the percentage of cells in the G0/G1 phase and accumulation of those in the S and G2/M phases of the cell cycle [23]. Collectively, these findings suggest that PdNPs can dose-dependently decrease cell viability and proliferation.

The mechanism of PdNPs induced toxicity in SKOV3 cells depends on the release of Pd ions into the media. The solubility of PdNPs and Pd ions play a crucial role in toxicity. Pd ions resulted as more toxic, compared to Pd-NPs at same concentrations (Figure 1C,D). The loss of cell viability and inhibition of cell proliferation due to the release of Pd ions released from Pd-NPs may have a role in their toxicological effects. Metal ions may accumulate in mitochondria damaging their function, which could induce mitochondrial-mediated apoptosis. However, deeper research is necessary to verify and confirm such possible mechanisms of action.

### 3.3. PdNPs Induce Leakage of LDH and Decrease Dead-Cell Protease Activity in SKOV3 Cells

Cytotoxicity can be assessed by measurement of the leakage of lactate dehydrogenase (LDH), a soluble molecule released into the surrounding extracellular space following cell exposure to cytotoxic compounds. When cell membrane integrity is compromised, detection of this enzyme in the culture medium can be used as a cell-death marker. Relative viability levels can be quantified by measuring the amount of released LDH, using a colorimetric LDH cytotoxicity assay. To evaluate LDH leakage, SKOV3 cells were treated with various PdNPs concentrations for 24 h, followed by measurement of LDH levels. The results indicated that LDH leakage is dose-dependent (Figure 3A).

Lactate dehydrogenase (LDH) leakage was significantly higher in treated cells and five-fold higher in the PdNPs-treated group compared to untreated. This finding was comparable with that in untreated control cells. Release of LDH is a cytotoxicity endpoint for measuring cell membrane integrity and viability [43]. Lactate dehydrogenase is one of the toxicity markers used to measure cell fate and indicate the stage of apoptosis/necrosis. Analysis of LDH leakage revealed that PdNPs significantly affect cell membrane integrity and induce cytotoxicity in SKOV3 cells. Among several biomarkers that have been described and employed for measuring viability and cytotoxicity in cell culture, we employed proteolytic biomarkers to determine the cytotoxic effects of PdNPs in the SKOV3 cell line. To further confirm the cytotoxic effect of PdNPs, dead-cell protease activity was measured using CytoTox-Glo, the most sensitive method available for measuring cytotoxicity. This assay quantifies the extracellular activity of an intracellular protease following its release from membrane-compromised cells [4]. To evaluate dead-cell protease activity, SKOV3 cells were treated with various PdNPs concentrations for 24 h. Dead cell protease activity was represented as cell viability. The results revealed that SKOV3 cell viability was significantly reduced at a PdNPs concentration of 5 µg/mL, and a reduction of 90% compared to untreated cells (Figure 3B). Collectively, both the LDH and dead-cell protease assays indicate that PdNPs induced significantly higher toxicity in SKOV3 cells in a dose-dependent manner.

### 3.4. PdNPs Induce Oxidative Stress and Increase MDA Levels

Oxidative stress is an imbalance between cellular oxidant production and antioxidant activity. Nanoparticles are known to initiate oxidative stress directly or indirectly through various mechanisms, thus exerting negative biological effects [44]. To verify the effects of PdNPs on oxidative stress, SKOV3 cells were treated with various PdNPs concentrations for 24 h. ROS levels in PdNPs-treated cells were then measured using enzymatic cleavage of DCFH-DA and expressed as a percentage of the untreated controls. The results showed that PdNPs dose-dependently induce ROS production in SKOV3 cells (Figure 4A). Previous reports have indicated that the effect of PdNPs on ROS production is limited exclusively to human ovarian cancer cells. For example, Wilkinson et al. demonstrated that PdNPs induce apoptosis dose-dependently in PBECs, but not A549 cells [19]. Moreover, we recently reported that plant extract-assisted PdNPs can increase ROS production in A2780 cells [14]. Excessive cellular ROS levels can induce oxidative stress, which impairs normal physiological redox-regulated functions. This, in turn, leads to DNA damage, unregulated cell signaling, changes in cell motility, cytotoxicity, and apoptosis [45]. Palladium NPs were shown to induce apoptosis, cytotoxicity, and DNA damage in human skin malignant melanoma (A375) cells by increasing oxidative stress [23].

Nanoparticles can interact directly with proteins, lipids, and DNA, leading to protein degradation, membrane damage, and mutagenesis. To examine whether the PdNPs-induced increase in ROS levels affects lipid peroxidation, SKOV3 cells were treated with various PdNPs concentrations for 24 h. Malondialdehyde is one of the end products of lipid peroxidation, and MDA levels can be used to determine the severity of lipid peroxidation in biological systems. The MDA levels were measured using the thiobarbituric acid reactive substances (TBARS) assay, a traditional and well-established method for detecting MDA levels. The results showed that treating SKOV3 cells with PdNPs increased intracellular MDA levels in a concentration-dependent manner, resulting in lipid peroxidation (Figure 4B). Dose-dependent ROS generation, as well as lipid peroxidation, have also been observed in human umbilical vein endothelial cells (HUVECs) treated with silicon NPs (SiNPs), indicated by increased intracellular levels of MDA [46]. Similarly, human bone marrow mesenchymal stromal cells treated with superparamagnetic iron oxide NPs (SPIONs) displayed lipid-related oxidative damage [47]. Previous studies have suggested that a combination of PdNPs and trichostatin A increases toxicity in human cervical cancer cells and that PdNPs, in combination with tubastatin-A, show increased toxicity in human breast cancer cells by increasing MDA levels [48,49]. Exposure of A375 cells to different PdNPs doses (0–40 μg/mL) significantly increased intracellular MDA levels dose-dependently, indicating that PdNPs may potentially induce oxidative damage [23]. Thus, our observations in the present study are in agreement with the findings of Alarifi et al. in A375 cells [23]. Collectively, the results indicate that lipid peroxidation is crucial for the PdNPs-induced toxicity in cancer cells.

### 3.5. Effect of PdNPs on Antioxidants

Antioxidant mechanisms are critical for the maintenance of oxidative balance. For example, several antioxidants play significant roles in oxidative balance through both enzymatic and non-enzymatic redox reactions, involving glutathione (GSH), superoxide dismutase (SOD), glutathione peroxidase (GPx), and vitamins [44]. These mechanisms help reduce the effects of pro-oxidants in biological systems. We therefore estimated the levels of two representative antioxidants, glutathione (GSH) and superoxide dismutase (SOD), in the presence of PdNPs. To determine the antioxidant status, SKOV3 cells were treated with various PdNPs concentrations for 24 h. The results indicated that PdNPs exposure leads to a dose-dependent reduction in the levels of antioxidants like GSH and SOD (Figure 5A,B). Mechanistically, the degree of oxidative stress depends on ROS levels and the presence of antioxidant defenses. Our findings suggest that PdNPs greatly reduce antioxidant levels, which is a major cause of toxicity in SKOV3 cells exposed to PdNPs. Alarifi et al. found reduced GSH levels at high PdNPs concentrations [23]. Regulating GSH levels is important for modulating oxidative stress, and GSH has emerged as a key endogenous antioxidant based on its cellular abundance, especially in the mitochondria [50].

### 3.6. PdNPs Induce Mitochondrial Dysfunction and Reduce ATP Generation

Mitochondria are a source of ROS, which are by-products of biochemical reactions like mitochondrial respiration and cytochrome P450 enzymatic metabolism. The mitochondrial membrane potential (MMP) is a key indicator of membrane integrity and reflects the pumping of hydrogen ions across the inner membrane during electron transport and oxidative phosphorylation processes, both of which are the driving forces behind ATP production [51]. A change in mitochondrial membrane permeability is one of the critical events in apoptosis. Therefore, the effect of PdNPs on the MMP in SKOV3 cells was evaluated using the JC-1 MMP detection kit. As shown in Figure 6A, compared to the control, treating SKOV3 cells with PdNPs resulted in a dose-dependent decrease in the JC-1 aggregate to monomer ratio. A significant difference was observed between PdNPs-treated and untreated cells. Similarly, PdNPs have also been shown to induce mitochondrial dysfunction in A2780 cells. Exposure to other NPs, such as SiO_2_NPs, results in impaired mitochondrial energy metabolism in hepatocytes through the interruption of mitochondrial membrane integrity [52]. Chen et al. reported that titanium dioxide NPs (TiO_2_NPs) caused significant mitochondrial dysfunction by increasing ROS levels, decreasing ATP generation, and decreasing the metabolic flux in the tricarboxylic acid (TCA) cycle of macrophages [51]. These findings indicate that PdNPs induce mitochondrial depolarization.

To provide further evidence for mitochondrial dysfunction following PdNPs exposure, we measured the ATP levels to analyze the functional aspects of mitochondria. Since mitochondria are the powerhouse of the cell, ATP is produced predominantly in mitochondria and the ATP level is a sensitive measure of mitochondrial function. We found that PdNPs trigger mitochondrial dysfunction, corroborating accumulating evidence that NPs target mitochondria after cell entry. As shown in Figure 6B, the ATP levels were dose-dependently attenuated in SKOV3 cells after incubation with PdNPs. Chen et al. demonstrated that TiO_2_NPs evoke significant mitochondrial dysfunction by increasing the levels of mitochondrial ROS and decreasing the generation of ATP and the metabolic flux in the tricarboxylic acid (TCA) cycle, both in RAW 264.7 cells and in bone marrow-derived macrophages [51]. Our findings suggest that mitochondrial function is markedly affected by a significant decrease in ATP production, likely through increased ROS generation, in the mitochondria of PdNPs-treated cells. Collectively, our studies provide strong evidence that exposing human ovarian cancer cells to PdNPs may elicit adverse effects in cellular responses and cytotoxicity.

### 3.7. PdNPs Induce DNA Damage and Caspase 3 Activation

DNA damage and DNA fragmentation are both important and irreversible events in apoptosis. To evaluate the involvement of ROS in PdNPs-induced damage in SKOV3 cells, 8-oxo-dG levels were determined by ELISA. After 24 h of exposure of SKOV3 cells to various PdNPs concentrations, increased oxidative DNA damage was indicated by significantly elevated levels of 8-oxo-dG production that was found to be concentration-dependent (Figure 7A). Exposure to PdNPs increases ROS generation, which can damage the lipid components of cellular membranes, proteins, and DNA. 8-Oxo-dG is one of the predominant free radical-induced products of DNA oxidation and has, therefore, been widely used as a biomarker for oxidative stress. The biomarker 8-OHdG or 8-oxo-dG has been a pivotal marker for measuring the effect of endogenous oxidative damage to DNA. Iavicoli et al. observed that PdNPs can induce a significant increase in DNA breaks within 4 h of incubation. Surprisingly, the intensity of DNA damage was severe in A549 cells after 72 h of exposure [42]. Additionally, the authors also found maintenance of the G0 state or prolongation/arrest in the G1 phase, which could indicate DNA damage that may prevent cell entry into the S phase of the cell cycle [42]. A possible reason for the PdNPs-induced DNA damage in SKOV3 cells is the ability of PdNPs to cross the cell membrane and enter the nucleus. In turn, this could activate a response to DNA damage by cell cycle inhibitors, such as CDKN1A and CDKN1B, and arrest of cell cycle progression in the G0/G1 phase [42]. Collectively, our findings suggest that PdNPs induce DNA damage, potentially due to excessive oxidative stress. This indicates that ROS are a major factor in PdNPs-induced cytotoxicity and elicit genotoxic effects, either through direct interaction with DNA or indirectly via NP-induced oxidative stress and apoptotic responses, in SKOV3 cells.

Caspases are crucial mediators of programmed cell death. Among the caspases, caspase 3 is typically involved in activating death proteases that catalyze the specific cleavage of many key cellular proteins [53]. Caspase 3 is triggered by several death signals, cleaving a number of cellular proteins essential for inducing DNA damage and morphological alterations in cells undergoing apoptosis [54]. We therefore determined whether PdNPs-induced caspase 3 activation elicits DNA damage. To examine PdNPs-induced cell death in SKOV3 cells, we quantified caspase 3 levels in PdNPs-treated cells through the cleavage of the caspase 3 substrate I (N-acetyl-DEVD-p-nitroaniline). The results suggest that caspase 3 activity increased in a dose-dependent manner following PdNPs exposure (Figure 7B), similar to that reported for caspase 3 activity in A375 cells [23]. Takaki et al. found that PdNPs induced apoptosis in leukemia L1210 cells by TiO_2_NPs was associated with chromosomal DNA fragmentation and caspase 3 activation [55]. Together, the results of the present study revealed that the PdNPs-induced cytotoxic and genotoxic effects, as well as the mode of death of SKOV3 cells, was mediated by ROS-triggered caspase 3 cleavage.

### 3.8. Effect of PdNPs on Cell Cycle

The consequences of DNA damage and caspase-3 activation, cells were leads to apoptosis. Next, we examined the effect of PdNPs on cell cycle analysis. The cells were treated with IC50 concentration of PdNPs and then we examined the cell cycle (Figure 8) and confirmed the quiescent/resting state of PdNPs-treated cells. The exposure of SKOV3 to 2.5 µg/mL PdNPs exhibited significant reduction of cells synthesizing DNA (S-phase) (34.0 ± 1.5% versus 45.7 ± 2.6%) a significant increase of cells within the G0/G1-phase (58.1 ± 1.8%versus 42.1 ± 2.0%), and a significant reduction of cells in G2/M-phase (4.5 ± 0.1% versus 8.0 ± 1.0%), compared to control cells. A unique pattern was observed in for Pd(IV) ion exposed cells, with significantly higher G0/G1 accumulation compared to control (71.5 ± 2.4% versus 42.1 ± 2.0%), lower percent of cells in S-phase (24.0 ± 1.1% versus 45.7 ± 12.6%), and a significant reduction of cells in G2/M-phase (0.8 ± 0.2% versus 8.0 ± 1.0%), compared with control cells (Figure 8). Interestingly, PdNPs and Pd(IV) ion-exposed cells are comparable with doxorubicin-treated SKOV3 cells. Doxorubicin-treated SKOV3 cells showed significantly higher G0/G1 accumulation compared to unexposed cells (60.2 ± 2.2% versus 42.1 ± 2.0%), lower percent of cells in S-phase (35.0 ± 1.1% versus 45.7 ± 12.6%), and a significant reduction of cells in G2/M-phase (1.3 ± 0.2% versus 8.0 ± 1.0%), compared with control cells (Figure 8). Similarly, Petrarca et al. [20] reported that PdNPs significantly disturbed cell cycle associated with a significant increase of cells within the G0/G1-phase and a significant reduction in GS- and G2/M-phases. Collectively, all these results suggest that PdNPs seems to be potential inducers of cytotoxicity in SKOV3 cells. All cell cycle data are summarized in (Table 2).

Previous studies and findings from this study concluded that Pd ions, released from NPs, could be the inducers of Pd toxicity [14,20]. Gurunathan et al. [14] demonstrated that human ovarian cancer cells exposed to PdNPs cause more marked subcellular alterations, including the presence of numerous auto-phagosomal vacuoles containing damaged mitochondria, and/or undigested cytoplasmic material, and a significant amplification of cell cycle alterations described for PdNPs [14,20]. Collectively, all these findings suggest that Pd ions released from PdNPs may have a role in their toxicological effect on human ovarian cancer cells. Furthermore, metal ions may accumulate in mitochondria damaging their function, which may in turn arrest the cell cycle and cause cell death [14,20]. However, significant studies are required to address the molecular mechanisms and cellular pathways involved in the mechanism of action of PdNPs.

### 3.9. PdNPs Exposure Alters the Expression Patterns of Multiple Genes in SKOV3 Cells

The precise regulation of epigenetic dynamics safeguards gene transcription, lineage commitment, and cell survival. Such epigenetic mechanisms include regulation of chromatin structure by ATP-dependent chromatin remodeling, histone modifications, DNA methylation, and non-coding RNA. Emerging evidence suggests that exposure to nanomaterials changes the status of DNA methylation and histone modification in cells through unknown mechanisms [56,57,58], leading to changes in gene expression and affecting cell viability. Therefore, we sought to explore the mechanistic link between PdNPs-mediated apoptosis and changes in gene expression by epigenetic regulation. To this end, we performed an RNA-Seq analysis of SKOV3 cells treated with PdNPs and generated more than 20 million reads/sample from two biological replicates/group. As illustrated in Figure 9A, 247 and 811 genes were down- and upregulated respectively, in PdNPs-treated cells. The expression patterns of representative genes (*SMYD3* and *HIST1H2BD* as downregulated and *DDIT3* as upregulated) were visualized in the Integrative Genomics Viewer (IGV) genome browser (Figure 9B). Using all of the 1058 DEGs, we performed gene ontology (GO) term analysis to identify the biological processes dysregulated by PdNPs exposure and found that epigenetics-related GO terms such as nucleosome assembly, telomere organization, and chromatin silencing at rDNA were significantly enriched (Figure 9C).

### 3.10. Genes Involved in Epigenetic Regulation and Apoptosis Were Aberrantly Expressed after PdNPs Exposure

Next, each down- or upregulated gene was subjected to biological process analysis using DAVID. The GO term nucleosome assembly was identified as the top-ranked biological pathway when the genes downregulated by PdNPs were subjected to DAVID analysis (Figure 10A). Other GO terms associated with epigenetics, including chromatin silencing at rDNA and positive regulation of gene expression, were also found among the downregulated genes. Interestingly, most genes in the GO term of nucleosome assembly were associated with histone gene clusters. Heatmaps showing the expression patterns of genes in the chromatin silencing at rDNA and nucleosome assembly GO terms are illustrated in Figure 10B.

Consistent with our data, exposure of *Arabidopsis thaliana* to NPs such as zinc oxide, fullerene soot, or titanium dioxide silences the expression of histone genes, as well as that of other genes related to nucleosome assembly [59]. Silencing of the histone H3K4 methyltransferase gene, *SMYD3*, a gene repressed by PdNPs, was shown to induce cell cycle arrest and apoptosis [60]. In contrast, GO terms related to transcription, development, cell death, and apoptosis were found for the genes upregulated by PdNPs (Figure 10C). The expression of *DDIT3*, a gene upregulated in the positive regulation of transcription from RNA polymerase II promoter GO term, was found to be enhanced in response to ER stress and also to be involved in apoptosis [61]. Heatmaps showing the expression patterns of genes in the positive regulation of transcription from RNA polymerase II promoter and regulation of cell death are listed in Figure 10D.

### 3.11. Treatment with PdNPs-Altered Biological Pathways Involved in Human Disease and Apoptosis

We next performed a detailed pathway analysis using Kyoto Encyclopedia of Genes and Genomes (KEGG), gene set enrichment analysis (GSEA), and GeneMANIA. As shown in Figure 11A, KEGG analysis revealed that pathways related to human diseases, including Huntington’s disease, Parkinson’s disease, Alzheimer’s disease, and systemic lupus erythematosus were impaired by PdNPs treatment. The data is intriguing as multiple neuronal disease-related pathways are dysregulated in the cell types derived from ovarian cancer cells. In addition, most genes involved in the MAPK signaling pathway were enhanced (indicated in red), whereas those involved in mitochondrial oxidative phosphorylation were suppressed (indicated in blue), consistent with our cellular assays indicating that the ATP level decreased significantly following PdNPs treatment. Previous studies showed that aberrant regulation of the MAPK signaling pathway or of oxidative phosphorylation induces apoptosis in various cell types [14,33]. The GSEA consistently revealed that genes involved in apoptosis and DNA repair were enriched among the differentially expressed genes (DEGs) (Figure 11B). Next, we analyzed both gene co-expression and physical interaction among elements involved in the MAPK signaling pathway and oxidative phosphorylation (Figure 11C). As expected, many of the genes were co-expressed and some proteins involved in oxidative phosphorylation, including CYC1-UQCR10, ATP5F1C- ATP5PB, and TNFSF11-ATP5MC1, interacted directly. Some proteins in the MAPK signaling pathway (HSPA1A, HSPA1B, HSPA2, and HSPA6) involved in apoptosis also interacted and were co-expressed in the same tissue or cell.

### 3.12. Transcription Factors Associated with Apoptosis Were Aberrantly Enhanced by PdNPs

Additional downstream analysis was performed to investigate whether the transcriptional change that induced apoptosis resulted from a direct regulatory effect elicited by PdNPs treatment. Differentially expressed transcription factors (TFs) were identified using the transcription factor database (TFDB) (Figure 12A). Eighty TFs and seven TFs were identified as upregulated and downregulated genes, respectively. No significant KEGG pathway was identified in the downregulated TF genes. In contrast, the 80 upregulated TFs were found in the myeloid leukemia, MAPK signaling, and Wnt signaling pathways (Figure 12B). More importantly, the expression of multiple genes downstream of the TFs, including *RUNX1*, *CEBPA*, *TCF7L1*, *JUND*, *DDIT3*, *NR4A1*, were also incorrectly regulated following PdNPs exposure (Figure 12C). Therefore, our analysis provides evidence of an association between PdNPs exposure and epigenetic transcriptional control and establishes a molecular basis for NP-mediated apoptosis.

## 4. Conclusions

Palladium NPs have attracted considerable interest recently due to their increased usage in catalytic processes, electronics, and nanomedicine. However, unlike for other metal NPs, there are no published reports on the cytotoxic potential of PdNPs, or on the PdNPs-associated cellular pathways and molecular mechanisms involved in cancer, particularly in SKOV3 cells. Therefore, this study aimed to investigate the cytotoxic potential of PdNPs and the molecular mechanism of apoptosis induced by PdNPs in SKOV3 cells. Several endpoints such as cell viability, cell proliferation, cytotoxicity, oxidative stress, mitochondrial dysfunction, DNA damage, caspase 3 activation, and induction of apoptosis were evaluated. The results showed that PdNPs inhibited cell growth and proliferation in a dose-dependent manner in SKOV3 cells. Palladium NPs-caused concentration-dependent cytotoxicity as determined by LDH leakage, increased ROS and MDA levels, and decreased antioxidant levels, including those of GSH and SOD. Furthermore, this study revealed that PdNPs induce mitochondrial dysfunction by altering the MMP and reducing ATP levels, DNA damage, and caspase 3 activation. Concerning possible modes of action of PdNPs to induce cytotoxicity in human ovarian cancer cells, oxidative stress is considered as one of the most crucial mechanisms for cytotoxic effects of NPs. In addition, lipoperoxidation, mitochondria-mediated apoptosis, involved in PdNPs induced apoptosis (Figure 13). This in vitro toxicity study provided important insights on the possibility of relevant consequences at the cellular level after exposure to PdNPs. The RNA-Seq data demonstrated that DEGs were enriched in various biological processes, including nucleosome assembly, telomere organization, and chromatin silencing. Additionally, GO term analysis also identified nucleosome assembly as the top-ranked biological pathway. Furthermore, our RNA-Seq analysis provided evidence for an association between PdNPs exposure and epigenetic transcriptional control and establishes a molecular basis for NP-mediated apoptosis. These findings provide valuable insights into the understanding of the molecular mechanisms responsible for PdNPs-induced cytotoxicity and may be useful in elucidating the mechanism underlying the effect of PdNPs on SKOV3 cells and for identifying novel targets for ovarian cancer treatment. 

## Figures and Tables

**Figure 1 nanomaterials-09-00787-f001:**
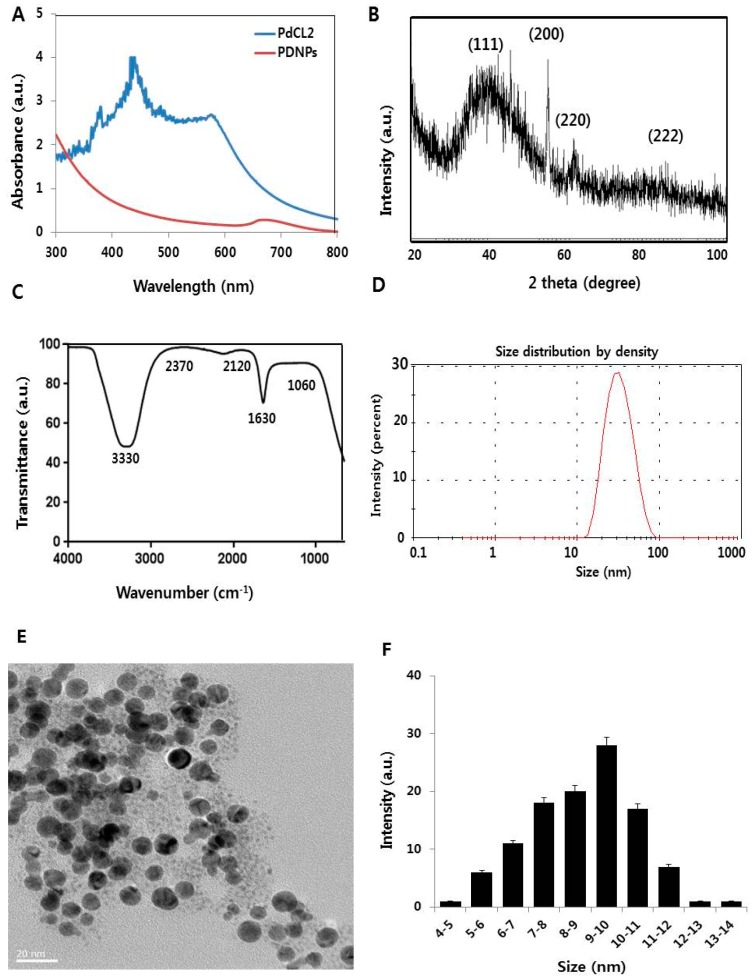
Synthesis and characterization of palladium nanoparticles (PdNPs). (**A**) Ultraviolet-visible spectra of PdCl_2_ (blue) and PdNPs (red), and (**B**) X-ray diffraction (XRD) pattern of PdNPs. (**C**) Fourier-transformed infrared (FTIR) spectra of PdNPs. (**D**) Size distribution analysis of PdNPs by dynamic light scattering (DLS). (**E**) Transmission electron microscopy (TEM) images of PdNPs. (**F**) Size distribution based on TEM images of PdNPs, ranging from 4 nm to 14 nm.

**Figure 2 nanomaterials-09-00787-f002:**
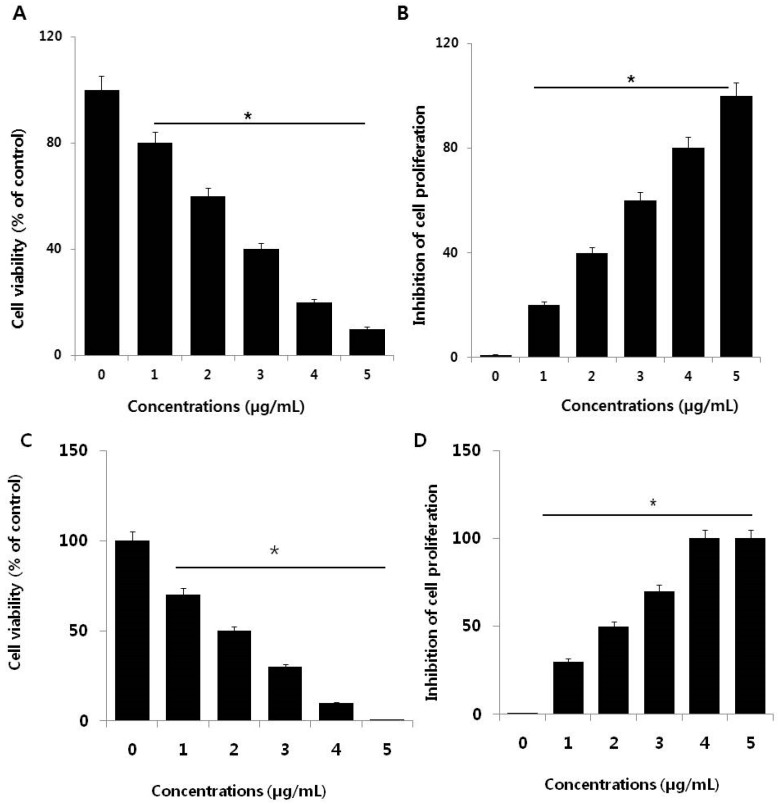
Effect of palladium nanoparticles (PdNPs) on SKOV3 cell viability and proliferation. (**A**) Viability of SKOV3 cells was determined by CCK-8 assay after a 24-h exposure to different concentrations of PdNPs. (**B**) The cell proliferation assay was performed using BrdU incorporation. A significant difference was observed between treated and untreated cells. Effect of palladium ions on SKOV3 cell viability and proliferation. (**C**) Viability of SKOV3 cells was determined by CCK-8 assay after a 24-h exposure to different concentrations of Pd ions. (**D**) The cell proliferation assay was performed using BrdU incorporation. A significant difference was observed between treated and untreated cells.

**Figure 3 nanomaterials-09-00787-f003:**
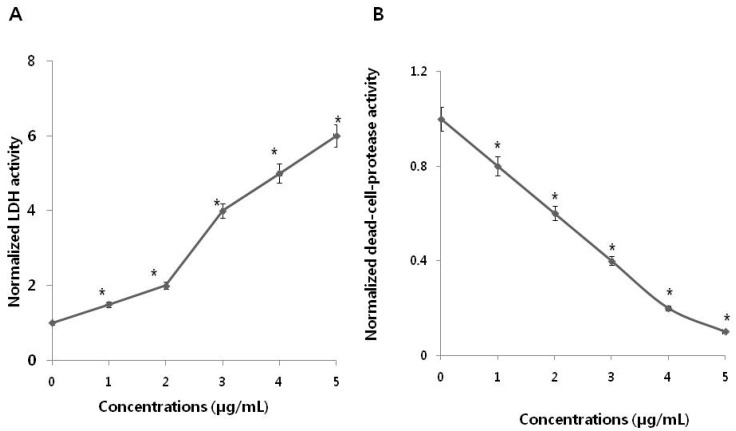
Measurement of lactate dehydrogenase (LDH) leakage and dead-cell protease activity in SKOV3 cells. (**A**) After 24 h of exposure to different PdNPs concentrations, LDH activity was measured at 490 nm, using the LDH cytotoxicity kit. Data were further normalized to LDH activity measured in cell medium to account for damaged/nonviable cells (normalized LDH activity). The PdNPs-pretreated, unexposed control LDH activity is equal to the value of 1 at each time point. These data are representative of three experiments. Error bars represent S.E.M. Asterisks (*) indicate statistically significant increase in LDH activity by treatment with PdNPs (* *p* < 0.05). (**B**) The levels of dead-cell protease were measured. There was a significant reduction in the number of PdNPs-treated cells compared to the number of untreated cells, as determined by the Student’s *t*-test (* *p* < 0.05). Data were further normalized to dead-cell protease measured in cell medium to account for damaged/nonviable cells (normalized dead-cell protease). The PdNPs-pretreated, unexposed control dead-cell protease is equal to the value of 1 at each time point. These data are representative of three experiments. Error bars represent S.E.M. Asterisks (*) indicate statistically significant increase in dead-cell protease activity by treatment with PdNPs (* *p* < 0.05).

**Figure 4 nanomaterials-09-00787-f004:**
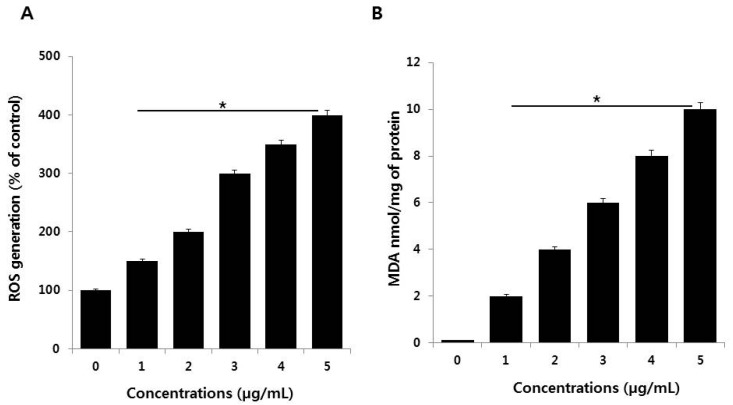
(**A**) Measurement of ROS (**B**) MDA levels in SKOV3 cells. There was a significant difference in the ratio for PdNPs-treated cells compared to that in untreated cells, as determined by the Student’s *t*-test (* *p* < 0.05).

**Figure 5 nanomaterials-09-00787-f005:**
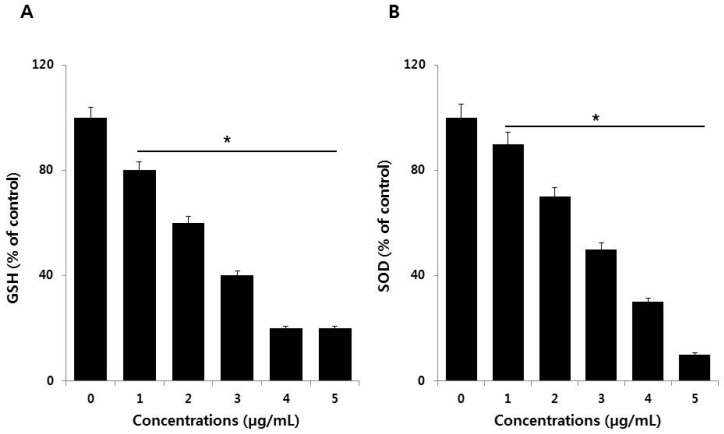
Measurement of antioxidant levels in SKOV3 cells. (**A**) The concentration of glutathione (GSH) is expressed as milligrams per gram of protein. (**B**) The specific activity of superoxide dismutase (SOD) is expressed as units per milligram of protein. Treated cells showed statistically significant differences compared to untreated cells, as determined by the Student’s *t*-test (* *p* < 0.05).

**Figure 6 nanomaterials-09-00787-f006:**
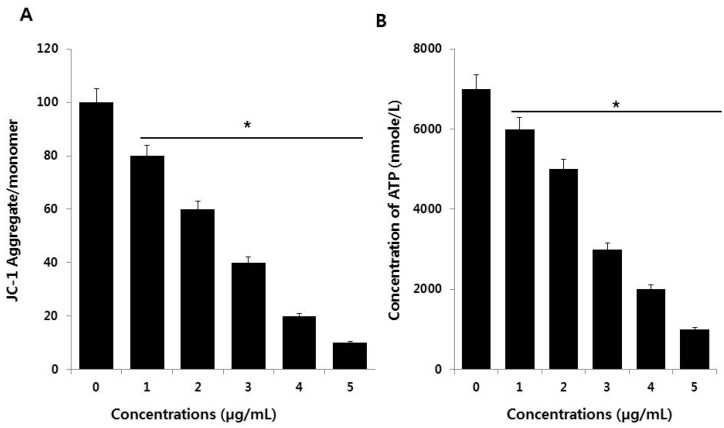
Measurement of mitochondrial membrane potential (MMP) (**A**) and ATP levels (**B**) in SKOV3 cells. There was a significant difference in the ratio in PdNPs-treated cells compared to that for in untreated cells, as determined by the Student’s *t*-test (* *p* < 0.05).

**Figure 7 nanomaterials-09-00787-f007:**
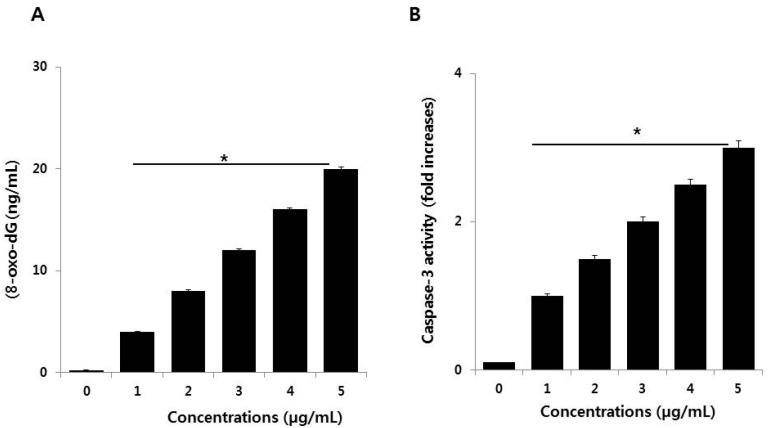
(**A**) Palladium nanoparticles (PdNPs) induce DNA damage and (**B**) activate caspase 3 in SKOV3 cells. The treated groups showed statistically significant differences compared to the control group, as determined by the Student’s *t*-test. (* *p* < 0.05).

**Figure 8 nanomaterials-09-00787-f008:**
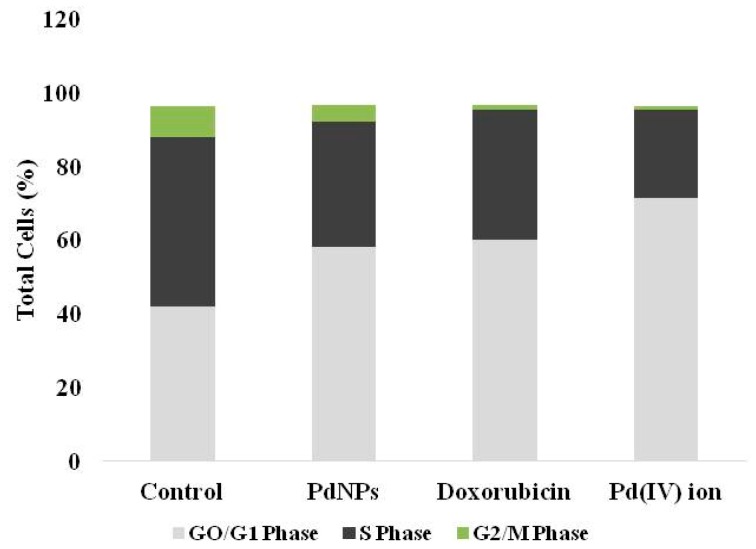
Cell cycle analysis of SKOV3 cells exposed to PdNPs, doxorubicin and Pd(IV) ion. The cells were treated with 2.5 µg/mL of PdNPs, doxorubicin, and Pd(IV) ion and for comparison, an equal volume of vehicle was used in parallel cultures. The proportion of cells in the different phases was quantitated and represented with partitioned bars as the average of three experiments. G0/G1-phase: light grey; S-phase: dark grey; G2/M phase: green.

**Figure 9 nanomaterials-09-00787-f009:**
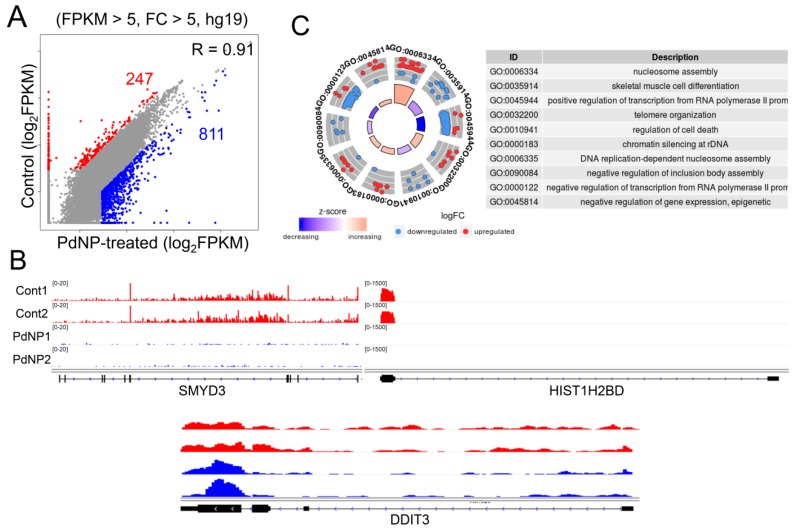
Palladium nanoparticle (PdNPs) exposure alters gene expression. (**A**) Scatter plot showing the downregulated (red dots) and upregulated (blue dots) genes after PdNPs exposure. Cutoff: FPKM > 5 and fold change (FC) > 5. (**B**) Integrative Genomics Viewer (IGV) images showing representative down- and upregulated genes. (**C**) GO-plot image of GO term analysis for all the differentially expressed genes (DEGs). Red dots and blue dots represent down- and upregulated genes, respectively. FPKM, fragments per kilobase of transcript, per million mapped reads.

**Figure 10 nanomaterials-09-00787-f010:**
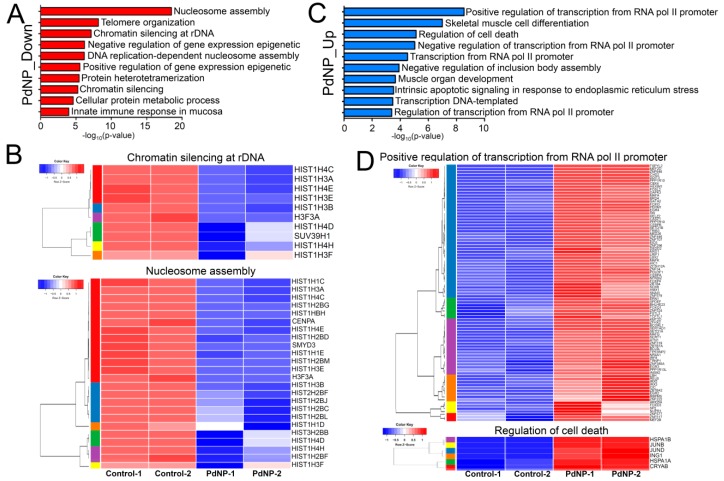
Genes involved in epigenetic and transcriptional regulation are significantly altered by PdNPs treatment. (**A**) Bar graphs showing GO terms enriched in downregulated genes. -log_10_ (*p*-value) is used. (**B**) Representative heatmaps showing genes in the chromatin silencing at ribosomal DNA and nucleosome assembly GO terms. Note that most of the genes in the GO terms are associated with histone gene clusters. (**C**) Bar graphs showing GO terms enriched in upregulated genes. (**D**) Representative heatmaps showing genes in the GO terms of positive regulation of transcription from RNA polymerase II promoter and regulation of cell death.

**Figure 11 nanomaterials-09-00787-f011:**
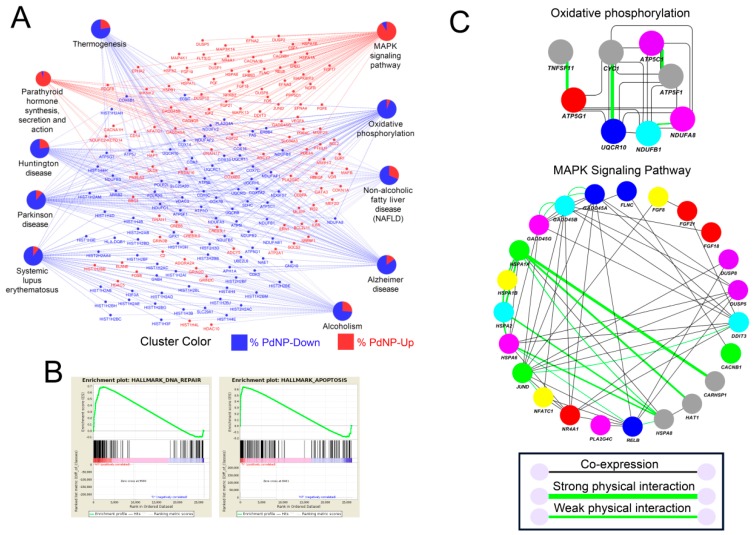
Pathway analysis reveals the dysregulated expression of apoptosis-inducing genes. (**A**) Kyoto Encyclopedia of Genes and Genomes (KEGG) analysis showing interconnected genes in the biological pathways. Red and blue colors represent genes respectively up- and downregulated by palladium nanoparticles (PdNPs). (**B**) Gene set enrichment analysis (GSEA) showing enriched pathways with both up- and downregulated genes. Note that DNA repair and apoptosis pathways were both detected within the differentially expressed genes (DEGs). (**C**) GeneMANIA analysis showing the gene co-expression network (grey lines) and protein–protein interactions (light green lines) in the KEGG pathways (oxidative phosphorylation and MAPK signaling pathway).

**Figure 12 nanomaterials-09-00787-f012:**
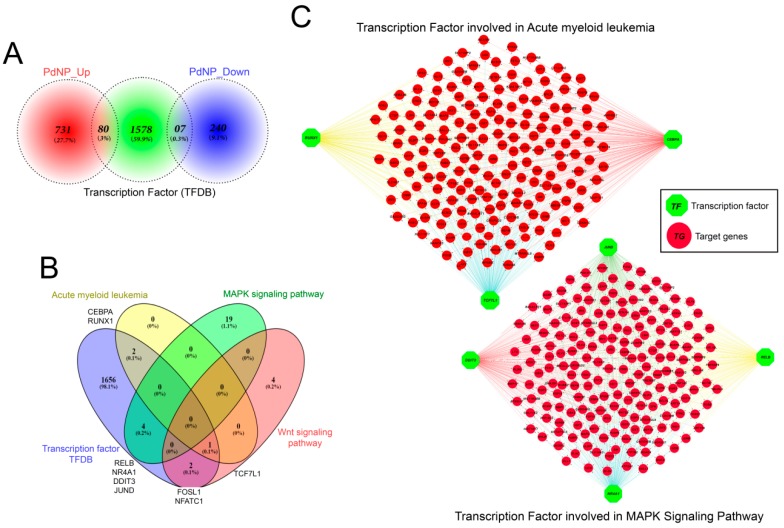
Discovery of critical transcription factors that directly regulate identified differentially expressed genes (DEGs) in palladium nanoparticle (PdNPs)-treated SKOV3 cells. (**A**) Ven diagram showing the number of transcription factors (TFs) among the DEGs. (**B**) Ven diagram showing the number of transcription factors identified when 80 upregulated TF genes were submitted to KEGG pathway analysis. Note that TFs acting in acute myeloid leukemia, MAPK signaling, and Wnt signaling pathways were significantly altered. (**C**) Gene network showing the interconnectedness of DEGs directly regulated by TFs such as *RUNX1*, *TCF7L1*, *CEBPA*, *JUND*, *DDIT3*, *RELB*, and *NR4A1*.

**Figure 13 nanomaterials-09-00787-f013:**
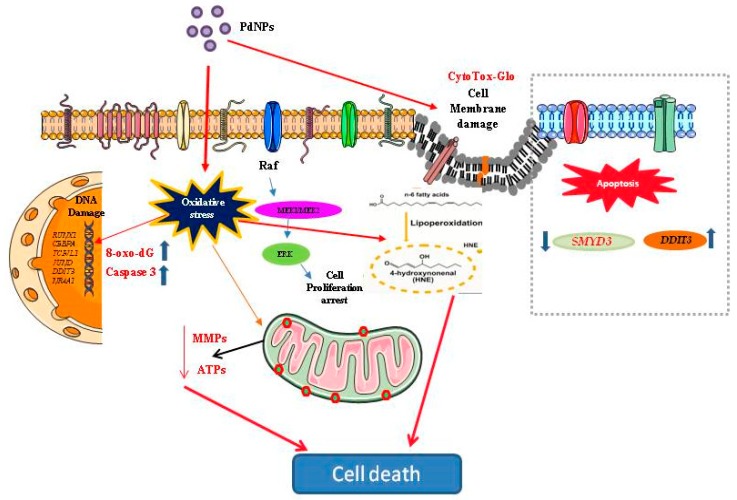
Hypothetical model demonstrating that the cellular mechanisms of PdNPs-induced oxidative stress, toxicity and gene expression.

**Table 1 nanomaterials-09-00787-t001:** Physicochemical characterization of PdNPs (mean ± SD, n = 3). Particle size and zeta potential in Dulbecco's Modified Eagle Medium (DMEM) were measured by ZetaSizer Nano (Malvern).

Name of the Sample PdNPs (Concentration (µg/mL))	Hydrodynamic Size in Dulbecco’s Modified Eagle’s Medium (DMEM)	Zeta Potential in Dulbecco’s Modified Eagle’s Medium (DMEM)
1	14	−15.4 ± 1.3
2	13	−17.3 ± 2.2
3	12	−18.8 ± 1.9
4	11	−19.5 ± 2.7
5	11	−21.8 ± 3.3

**Table 2 nanomaterials-09-00787-t002:** Cell cycle distribution data of SKOV3 cells exposed to PdNPs, or doxorubicin or Pd(IV) ion.

Phase	Control	PdNPs	Doxorubicin	Pd(IV) ion
GO/G1	42.1 ± 2.0	58.1 ± 1.8 *	60.2 ± 2.2 *	71.5 ± 2.4 *
S	45.7 ± 2.6	34.0 ± 1.5 *	35.1 ± 1.1 *	24 ± 1.1 *
G2/M	8.5 ± 0.5	4.5 ± 0.1 *	1.3 ± 0.1 *	0.8 ± 0.2 *

Data from three experiments are shown as mean ± S.D. * *p* < 0.05 PdNPs or Doxorubicin or Pd(IV) vs not exposed.

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
