# Peer review of "Cytotoxicity and Transcriptomic Analyses of Biogenic Palladium Nanoparticles in Human Ovarian Cancer Cells (SKOV3)"

_nanomaterials, 2019, doi:10.3390/nano9050787_

Reviewer 1 Report

Here the authors studied the effect of palladium NPs (PdNPs) and the molecular mechanisms and cellular pathways involved in ovarian cancer cell line via cell viability, proliferation, cytotoxicity, oxidative stress, DNA damage, apoptosis and RNA-Seq analysis. This study is interesting, however, some comments are listed below:

1.     Is PdNPs specific to SKOV3? Why choose this cell line or cancer cell type?

2.     What methord are used to test the cell survival and proliferation? CCK8 and /or Brdu? Have you calculated the IC50? The Method part are too simple.

3.     Line 206 “2.3. PdNPs induce toxicity in SKOV3 cells” is not accurate, LDH cytotoxicity assay was used for relative viability levels in this study, is that for toxicity assessment?

   5.  Line graphs could be used instead of some of the histograms through out the manuscript.

Author Response

Response to the reviewer comments-1

We immensely thank the reviewer valuable and constructive comments that greatly facilitated us for improving the overall quality of the manuscript. As per the reviewers’ constructive comments, the corrections were carried out in the manuscript. We hopefully believe that we have addressed all the comments mentioned by the reviewers carefully and precisely. All the changes are highlighted in yellow color in the revised manuscript. In addition, this manuscript was proof read by native English speaker by Editage editing company, Seoul, South Korea.

Comments and Suggestions for Authors

Here the authors studied the effect of palladium NPs (PdNPs) and the molecular mechanisms and cellular pathways involved in ovarian cancer cell line via cell viability, proliferation, cytotoxicity, oxidative stress, DNA damage, apoptosis and RNA-Seq analysis. This study is interesting, however, some comments are listed below:

Thanks to the reviewer for encouraging, positive and constructive comments to improve overall quality of the manuscript.

1.     Is PdNPs specific to SKOV3? Why choose this cell line or cancer cell type?

Thanks to the reviewer for excellent question. Response to your first question, there is no specificity between PdNPs and SKOV3. The reason for selection of SKOV3 as follows

Response to your second question, recently Beaufort et al (2014) characterized 39 ovarian cancer cell lines under uniform conditions for growth characteristics, mRNA/microRNA expression, exon sequencing, drug response for clinically-relevant therapeutics and collated all available information on the original clinical features and site of origin. Of the 39 ovarian cancer cell lines, 14 were assigned as high-grade serous, four serous-type, one low-grade serous and 20 non-serous type. SKOV3 comes under high grade serous and clinically important cell lines, therefore we selected to study the effect of PdNPs on SKOV3 cells.

2.     What method are used to test the cell survival and proliferation? CCK8 and /or Brdu? Have you calculated the IC50? The Method part are too simple.

Thanks to the reviewer for reminding missing information. Sorry for the typographical error. Cell viability was measured using a Cell Counting Kit-8 (CCK-8; CK04-01, Dojindo Laboratories, Kumamoto, Japan). Cell proliferation was determined using BrdU according to the manufacturer’s instructions (Roche).

Response to your second question, we calculated IC50. Concentrations of PdNPs showing a 50% reduction in cell viability (ie, half-maximal inhibitory concentration [IC50] values) were then calculated. The IC50 was found to be 2.5 µg/mL. RNA-SEQ analysis was carried out with the IC50 value.

3.     Line 206 “2.3. PdNPs induce toxicity in SKOV3 cells” is not accurate, LDH cytotoxicity assay was used for relative viability levels in this study, is that for toxicity assessment?

Thanks to the reviewer for thought-provoking comments. We absolutely agree with reviewer comments. According to the reviewer comments we changed section 2.3, PdNPs induce toxicity in SKOV3 cells into PdNPs induce leakage of LDH and decrease dead-cell protease activity in SKOV3 cells.

In our experimental setup LDH assay was used to measure cytotoxicity in SKOV3 cells. We didn’t use to measure relative cell viability. Relative cell viability as well as  cytotoxicity was measured by another assay called dead cell protease activity. Generally LDH assay considered to be significant biomarker for toxicity studies and also to measure plasma membrane integrity.

4. Line graphs could be used instead of some of the histograms through out the manuscript.

Thanks to the reviewer for logical idea. According to the reviewer suggestions we tried to change only figure 3, when we placed line graph instead of histogram, the line graphs are not looking very nicely compared to histogram and histograms seems to be better for readers. Therefore we placed histogram in other figures. I hope the reviewer could understand our presentation also.

Once again thanks to the reviewer for wonderful comments to improve the overall quality of the manuscript.

Reviewer 2 Report

The manuscript by Gurunathan et al. reports the effects of "green" palladium nanoparticles on SKOV3 ovarian cancer cells. Authors should be commended for the rich amount of data presented, which have been obtained with a wide array of techniques, some of which fairly advanced. However, I have severla criticisms.

1) My most important concern is that no definite model emerges from the tens of graphs and diagrams. Can the Authors summarize their view on mechanisms underlying PdNP toxicity in a definite model, open to experimental validation? What comes before and what after?

2) I think that this limitation is a direct consequence of the fact that only one, fairly advanced, time point is chosen (24h). At this time point, cell viability is already heavily compromised, thus preventing any real possibility to investigate toxcity mechanisms. Indeed, although Authors do not provide any IC50 value, it seems likely that this value is fairly constant whatever endpoint is investigated (from Figure1 to Figure 7). This seems quite non-specific.

3) Besides these general considerations, some speific concerns must be also raised. For instance, no information on the possibility that Pd is released in the medium is given, so that the reader cannot understand if the effects reported are due to NP or Pd ions released. A proper control of this issue would be compare the effects of NP with those of equivalent doses (in terms of the metal) of Pd ions.

4) Several ions (e.g. copper and silver) interfere with several biochemical tests of viability/toxicity/enzyme activities. However, no control of this issue is given throughout the manuscript. Moreover, data on LDH activity should be given relative to a "100%" toxicity produced by the lysis of all the cells in the population (usially achieved with detergents). The fact that LDH activity is four-fold increased compared to control is not per se very informative (Fig. 3A).

5) While the decrease in GSH is expected in cells undergoing oxidative stress, the apparent sppression of SOD activity is quite strange since a compensatory induction of the enzyme should be expected. This result seems simply a consequence of the cell death rather than a mechanism invlved in its pathogenesis.

6) Some sentences are quite unclear. For instance, lines 221-222. LDH activity in treated cells is higher than control or comparable to control? Some terms are also ambiguously used: for instance, do cells die through necrosis (as suggested by LDH changes) or apoptosis (as implied by caspase results)?

7) RNASeq analysis is a very powerful technique still relatively novel in nanotoxicology and the information provided would be of interest. However, how much the changes in gene expression reported are a direct consequence of PdNP treatment or, rather, the effect of the advanced death process. Incidentally, what was the dose of PdNP exploited in that experiment?

8) Statistics. The description provided in the methods would sugest that data shown are taken from single, representative experiments, although authors state that at least three experiments have been performed for each endpoint. Please clarify.

9) Finally, possible conclusions on the therapeutic potential of PdNPs in ovarian cancer are absolutely overreaching, since, at least, they would require the use of more than a single cell line and an attempt to detect possible selective toxicity through a proper comparison with normal cell models.   

Author Response

Response to the reviewer comments-2

We immensely thank the reviewer valuable and constructive comments that greatly facilitated us for improving the overall quality of the manuscript. As per the reviewers’ constructive comments, the corrections were carried out in the manuscript. We hopefully believe that we have addressed all the comments mentioned by the reviewers carefully and precisely. All the changes are highlighted in yellow color in the revised manuscript. In addition, this manuscript was proof read by native English speaker by Editage editing company, Seoul, South Korea.

Comments and Suggestions for Authors

The manuscript by Gurunathan et al. reports the effects of "green" palladium nanoparticles on SKOV3 ovarian cancer cells. Authors should be commended for the rich amount of data presented, which have been obtained with a wide array of techniques, some of which fairly advanced. However, I have several criticisms.

Thanks to the reviewer for encouraging, positive and constructive comments to improve overall quality of the manuscript.

1) My most important concern is that no definite model emerges from the tens of graphs and diagrams. Can the Authors summarize their view on mechanisms underlying PdNP toxicity in a definite model, open to experimental validation? What comes before and what after?

Thanks to the reviewer for conclusive idea. According to the reviewer comments we summarize all the findings as a model describes the importance of findings, mechanism of toxicity caused by PdNPs to SKOV3 cells. The new figure included as 13 in the revised manuscript.

2) I think that this limitation is a direct consequence of the fact that only one, fairly advanced, time point is chosen (24h). At this time point, cell viability is already heavily compromised, thus preventing any real possibility to investigate toxcity mechanisms. Indeed, although Authors do not provide any IC50 value, it seems likely that this value is fairly constant whatever endpoint is investigated (from Figure1 to Figure 7). This seems quite non-specific.

Thanks to the reviewer for excellent and thought-provoking comments. We absolutely agree with reviewer, at 24 h the cell viability was heavily compromised. We calculated IC50 concentrations of PdNPs showing a 50% reduction in cell viability (ie, half-maximal inhibitory concentration [IC50] values) were then calculated. The IC50 was found to be 2.5 µg/mL. Instead of doing experiments with only one dose such as IC50 concentration, we used dose-dependent effect.

3) Besides these general considerations, some specific concerns must be also raised. For instance, no information on the possibility that Pd is released in the medium is given, so that the reader cannot understand if the effects reported are due to NP or Pd ions released. A proper control of this issue would be compare the effects of NP with those of equivalent doses (in terms of the metal) of Pd ions.

Thanks to the reviewer for critical evaluation of the manuscript. PdNPs induced toxicity is due to PdNPs. According to the reviewer suggestion, we performed new experiment on cell viability, cell proliferation and cell cycle analysis using PdNPs and Pd(IV) ion, which cause severe toxicity compared to PdNPs. The new figure was shown as Figure 1CD and figure 8 in the revised manuscript.

4) Several ions (e.g. copper and silver) interfere with several biochemical tests of viability/toxicity/enzyme activities. However, no control of this issue is given throughout the manuscript. Moreover, data on LDH activity should be given relative to a "100%" toxicity produced by the lysis of all the cells in the population (usially achieved with detergents). The fact that LDH activity is four-fold increased compared to control is not per se very informative (Fig. 3A).

Thanks to the reviewer for constructive and technically viable comments. According to the reviewer comments, we performed new experiments and presented both LDH and dead cell protease activity as relative.

5) While the decrease in GSH is expected in cells undergoing oxidative stress, the apparent sppression of SOD activity is quite strange since a compensatory induction of the enzyme should be expected. This result seems simply a consequence of the cell death rather than a mechanism involved in its pathogenesis.

Thanks to the reviewer for intellectual and technically viable question. Mostly nanoparticle treated cells exhibited decreased level of GSH, it is common phenomenon. We absolutely agree with reviewer about the suppression of SOD is a consequence of cell death or it could be due to threshold level of SOD is not able to tolerate the PdNPs induced toxicity.

However, our data concurrence with previous study reported that ZN-doped TiO2 nanoparticles suppress both GSH and SOD level in human breast cancer cells (Ahamed et al., 2016) and graphene oxide and reduced graphene oxide in TM3 and TM4 cells (Gurunathan et al., 2019).

6) Some sentences are quite unclear. For instance, lines 221-222. LDH activity in treated cells is higher than control or comparable to control? Some terms are also ambiguously used: for instance, do cells die through necrosis (as suggested by LDH changes) or apoptosis (as implied by caspase results)?

Thanks to the reviewer for keen observation of each and every sentences. According to the reviewer, the ambiguity sentences were rectified.

7) RNASeq analysis is a very powerful technique still relatively novel in nanotoxicology and the information provided would be of interest. However, how much the changes in gene expression reported are a direct consequence of PdNP treatment or, rather, the effect of the advanced death process. Incidentally, what was the dose of PdNP exploited in that experiment?

Thanks to the reviewer for excellent and remarkable question. As we expected your question in quite earlier while designing of our work we used only IC50 concentration (2.5 µg/mL) rather than higher concentration (5.0 µg/mL) to avoid any gene expression is due to cell death.

8) Statistics. The description provided in the methods would suggest that data shown are taken from single, representative experiments, although authors state that at least three experiments have been performed for each endpoint. Please clarify. 

We are extremely sorry for the confusion. We rectified the error in materials and method section. All the tests were conducted in triplicate, and the results are presented as the means ± standard deviation.

9) Finally, possible conclusions on the therapeutic potential of PdNPs in ovarian cancer are absolutely overreaching, since, at least, they would require the use of more than a single cell line and an attempt to detect possible selective toxicity through a proper comparison with normal cell models.  

Thanks to the reviewer for very interesting comment. Previously we analyzed the effect of PdNPs on embryonic fibroblast cells, the effect was not significant compared to cancer cells. The effectiveness of cancer cells is due to pH and other microenvironment factors of cancer cells.

In addition, our group and other research group already showed the effect of palladium nanoparticles in other cancer cell lines including human ovarian cancer cells (A2780), human breast cancer cells, and human cervical cancer cells. The interesting and significance of this story is to explain the comprehensive effect (cytotoxicity and transcriptomic analysis) of palladium nanoparticles on human ovarian cancer cells (SKOV3). Particularly, Beaufort et al (2014) characterized 39 ovarian cancer cell lines under uniform conditions for growth characteristics, mRNA/microRNA expression, exon sequencing, drug response for clinically-relevant therapeutics and collated all available information on the original clinical features and site of origin. Of the 39 ovarian cancer cell lines, 14 were assigned as high-grade serous, four serous-type, one low-grade serous and 20 non-serous type. SKOV3 comes under high grade serous and clinically important cell lines, therefore we selected to study the effect of PdNPs on SKOV3 cells.

 Once again thanks to the reviewer for wonderful and constructive comments to improve the overall quality of the manuscript.

Reviewer 3 Report

This paper describes cytotoxicity and transcriptomic analyses of biogenic palladium nanoparticles in human ovarian cancer cells.  This is a valuable report in nano-oncology. However, some points need clarifying and certain statements require further justification.  These are given below.

(1)As mentioned by the authors, this study is a first and valuable study of cellular pathways in ovarian cancer cells after PdNPs exposure. The authors should comment the cytotoxic mechanism, especially ROS and the cellular pathways detected in this study.  The authors should comment whether these pathways are targets of ovarian cancer therapy.  In addition, the authors should show what kinds of ovarian cancer therapy based on these results.

(2)The authors should show the Zeta potential and DLS in culture medium, and each concentration of PdNPs.

(3)The authors should show cell cycle data. Especially, cell arrest is important because the cell cycle arrest has been reported, and important for apoptosis.

(4)In this study, could the authors observe autophagy in ovarian cancer cells?

(5)Pd is a material characterized by high catalyst.  In medical use, how to use this PdNPs, the authors should comment.

Author Response

Response to the reviewer comments-3

We immensely thank the reviewer valuable and constructive comments that greatly facilitated us for improving the overall quality of the manuscript. As per the reviewers’ constructive comments, the corrections were carried out in the manuscript. We hopefully believe that we have addressed all the comments mentioned by the reviewers carefully and precisely. All the changes are highlighted in yellow color in the revised manuscript. In addition, this manuscript was proof read by native English speaker by Editage editing company, Seoul, South Korea.

Comments and Suggestions for Authors

This paper describes cytotoxicity and transcriptomic analyses of biogenic palladium nanoparticles in human ovarian cancer cells.  This is a valuable report in nano-oncology. However, some points need clarifying and certain statements require further justification.  These are given below.

Thanks to the reviewer for positive, encouraging and constructive comments to further improve the overall quality of the manuscript.

(1)As mentioned by the authors, this study is a first and valuable study of cellular pathways in ovarian cancer cells after PdNPs exposure. The authors should comment the cytotoxic mechanism, especially ROS and the cellular pathways detected in this study.  The authors should comment whether these pathways are targets of ovarian cancer therapy.  In addition, the authors should show what kinds of ovarian cancer therapy based on these results.

Thanks to the reviewer for excellent comments. Response to your first comment, we proposed model for cellular pathway modulated by PdNPs particularly oxidative stress, lipo-peroxidation, mitochondrial dysfunctions and DNA damage.  The new model figure was shown as figure 13 in the revised manuscript.

Response to your second comment, concerning possible modes of action of PdNPs to induce cytotoxicity in human ovarian cancer cells, oxidative stress is considered as one of the most crucial mechanisms for cytotoxic effects of PdNPs. In addition, lipoperoxidation, mitochondria mediated apoptosis involved in PdNPs induced apoptosis.

Response to your third comment, so far radiation, harmone and chemo therapy are frequently used for ovarian cancer. However all the existing therapies are having undesired side effects. To overcome these effects, nanoparticle mediated therapy or nanoparticle mediated combination therapy are viable. Palladium is a noble metal with remarkable catalytic, mechanic and electronic properties. Palladium nanostructures have gained interest in the last decade in a number of applications including catalysis. There is few studies are available in nanomedicine. In medicine, palladium is nowadays commonly used in dental appliances and Pd needles are used in the clinic for prostate cancer and choroidal melanoma brachytherapy. For example, polymer micelle nanotechnology provided therapeutic efficacy of anti-cancer drugs and minimizing the side effects. Recently porous Pd nanoplates exhibited exceptional all-round excellence in photothermal conversion, therapeutic gene loading/releasing, cytotoxicity, and in vitro combination cancer treatment. We believe that our findings also could broadens the potential applications of PdNPs in ovarian cancer therapy.  Particularly, PdNPs could be used in ovarian cancer therapy as anticancer agent, drug delivery agent, and intraperitoneal therapeutic agent.

(2)The authors should show the Zeta potential and DLS in culture medium, and each concentration of PdNPs.

Thanks to the reviewer for interesting comment. According to the reviewer suggestion we included Zeta potential and DLS of PdNPs in culture medium shown as Table 1 in the revised manuscript.

(3)The authors should show cell cycle data. Especially, cell arrest is important because the cell cycle arrest has been reported, and important for apoptosis.

Thanks to the reviewer for remarkable and constructive comments. According to the reviewer, we performed cell cycle analysis and the new data were included as figure 8 and all the data were summarized in table 2 in the revised manuscript.

(4)In this study, could the authors observe autophagy in ovarian cancer cells?

Thanks to the reviewer for interesting question. Previously, Petrarca et al and our group showed PdNPs could induce autophagy. Therefore we didn’t look at in this work. This work is mainly focused on comprehensive effect of PdNPs on cytotoxicity and transcriptome.

(5)Pd is a material characterized by high catalyst.  In medical use, how to use this PdNPs, the authors should comment.

Thanks to the reviewer for thought-provoking and intellectual comment. We absolutely agree with reviewer, yes palladium nanoparticles are high catalyst. Palladium nanostructures are characterized by remarkable catalytic and optical properties. However, until recently, very few studies have taken advantage of these unique characteristics for applications in the biomedical field. Very recently, palladium nanostructures have been reported as prodrug activator, as photothermal agents and for anti-cancer/anti-microbial agents.

For example by the use of their optical properties, PdNPs can be used for diagnosis or highly attractive theranostic purposes. The remarkable multiple facets of palladium nanostructures might be useful for combination therapy with anticancer drug.

Once again thanks to the reviewer for wonderful and constructive comments to improve the overall quality of the manuscript.

Round  2

Reviewer 1 Report

The authors answers well, I have no more question.

Author Response

Response to the reviewer comments-1-RII

Comments and Suggestions for Authors

The authors answers well, I have no more question.

We immensely thank the reviewer for valuable and constructive comments that greatly facilitated us for improving the overall quality of the manuscript. Once again thank you for your positive and encouraging comments.

Once again thanks to the editor and reviewer for wonderful and constructive comments to improve the overall quality of the manuscript.

Reviewer 2 Report

Authors have tried to address my remarks.

The manuscript has significantly improved.

However, lacking direct experimental data on NP cell uptake and/or release of Pd under the experimental conditions adopted (at least, I could not find them), a cautelative note should be added in the discussion about the possibility that, at least in part, the toxic effects could be due to Pd ions.  

Author Response

Response to the reviewer comments-2

Comments and Suggestions for Authors

Authors have tried to address my remarks.

The manuscript has significantly improved. However, lacking direct experimental data on NP cell uptake and/or release of Pd under the experimental conditions adopted (at least, I could not find them), a cautelative note should be added in the discussion about the possibility that, at least in part, the toxic effects could be due to Pd ions.

Response to your first comment, first of all we immensely thank the reviewer for valuable and constructive comments that greatly facilitated us for improving the overall quality of the manuscript. Once again thank you for your positive and encouraging comments.

Response to your second comment, we have shown the experimental data about release of Pd ion effect on cell cycle in figure 8 in the first revision and also we previously demonstrated that PdNPs uptake by human ovarian cancer cells (Gurunathan et al., 2015).

Previously we demonstrated the internalization of PdNPs and consequences in human ovarian cancer cells as follows

The cells treated with PdNPs exhibit internalization of PdNPs into the cytoplasm which induced the formation of numerous autophagosomes. Interestingly, the PdNPs did not only induce autophagosomes but were also localized in the autophagosomes and autolysosomes.

In addition, the PdNPs induced the formation of autolysosomes, which contain dead cell with electron-lucent cytoplasm and damaged organelles. Our previous results showed that the activation of autophagy was accompanied by an increase in ROS levels in cells exposed to PdNPs.

In addition, previous studies from Petrarcaet al (2014), Gurunathan et al (2015) and also findings from this study study concluded that Pd ions, released from NPs could be the inducers of Pd toxicity. Gurunathan et al demonstrated that human ovarian cancer cells exposed to PdNPs causes more marked subcellular alterations, including the presence of numerous auto-phagosomal vacuoles containing damaged mitochondria, and/or undigested cytoplasmic material, and a significant amplification of cell cycle alterations described for PdNPs. Collectively, all these findings suggest that Pd ions released from PdNPs may have a role in their toxicological effect on human ovarian cancer cells. Furthermore, metal ions may accumulate in mitochondria damaging their function which may in turn arrest the cell cycle and cause cell death. However, significant studies required to address molecular mechanism and cellular pathways involved in mechanism of action of PdNPs.

In order to avoid repetition of same experiments and also to demonstrate the comprehensive effect of PdNPs on cytotoxicity and tranascriptomic analyses, in this manuscript we mainly focused on cellular pathway analysis using various cellular assays and transcriptomic analysis by RNA-SEQ technology.

Finally according to the reviewer suggestion, a cumulative note was included in the discussion about the possibility that, at least in part, the toxic effects could be due to Pd ions in the page number 17 in the revised manuscript (RII).

Once again thanks to the editor and reviewer for wonderful and constructive comments to improve the overall quality of the manuscript.
